# COVID-19 Vaccination Intent and Willingness to Pay in Bangladesh: A Cross-Sectional Study

**DOI:** 10.3390/vaccines9050416

**Published:** 2021-04-21

**Authors:** Russell Kabir, Ilias Mahmud, Mohammad Tawfique Hossain Chowdhury, Divya Vinnakota, Shah Saif Jahan, Nazeeba Siddika, Samia Naz Isha, Sujan Kanti Nath, Ehsanul Hoque Apu

**Affiliations:** 1School of Allied Health, Faculty of Health, Education, Medicine and Social Care, Anglia Ruskin University, Chelmsford, Essex CM1 1SQ, UK; dv234@student.aru.ac.uk (D.V.); ssj151@student.aru.ac.uk (S.S.J.); 2Department of Public Health, College of Public Health and Health Informatics, Qassim University, Al Bukairiyah 52741, Saudi Arabia; i.emdadulhaque@qu.edu.sa; 3Department of Dental Public Health, Sapporo Dental College, Dhaka 1230, Bangladesh; tawfique@sdch.edu.bd (M.T.H.C.); knsujan@yahoo.com (S.K.N.); 4Center for Environmental and Respiratory Health Research (CERH), Faculty of Medicine, University of Oulu, 90014 Oulu, Finland; nazeeba.siddika@oulu.fi; 5CAPABLE-A Cambridge-Led Programme in Bangladesh, University of Cambridge, Cambridge CB2 1TN, UK; sni25@medschl.cam.ac.uk; 6The Intervention Centre, Oslo University Hospital, 0372 Oslo, Norway; hoqueapu@msu.edu; 7Department of Biomedical Engineering, Institute of Quantitative Health Science, Michigan State University, East Lansing, MI 48824, USA

**Keywords:** COVID-19 vaccine, health belief model, vaccination intent, Bangladesh

## Abstract

This article reports the intent to receive a SARS-COV-2 vaccine, its predictors and willingness to pay in Bangladesh. We carried out an online cross-sectional survey of 697 adults from the general population of Bangladesh in January 2021. A structured questionnaire was used to assess vaccination intent. The questionnaire included sociodemographic variables and health belief model constructs which may predict vaccination intent. Among the participants, 26% demonstrated a definite intent, 43% probable intent, 24% probable negative, and 7% a definite negative intention. Multivariable logistic regression analyses suggest an association between definite intent and previous COVID-19 infection (OR: 2.86; 95% CI: 1.71–4.78), perceiving COVID-19 as serious (OR: 1.93; 1.04–3.59), the belief that vaccination would make them feel less worried about catching COVID-19 (OR: 4.42; 2.25–8.68), and concerns about vaccine affordability (OR: 1.51; 1.01–2.25). Individuals afraid of the side effects (OR: 0.34; 0.21–0.53) and those who would take the vaccine if the vaccine were taken by many others (OR: 0.44; 0.29–0.67) are less likely to have a definite intent. A definite negative intent is associated with the concern that the vaccine may not be halal (OR: 2.03; 1.04–3.96). Furthermore, 68.4% are willing to pay for the vaccine. The median amount that they are willing to pay is USD 7.08. The study findings reveal that the definite intent to receive the SARS-CoV-2 vaccination among the general population varies depending on their COVID-19-related health beliefs and no significant association was found with sociodemographic variables.

## 1. Introduction

The coronavirus disease 2019 (COVID-19) is caused by the severe acute respiratory syndrome coronavirus 2 (SARS-Cov-2) [1]. The disease was first reported in Wuhan, China in December 2019 [2,3]. The disease then spread widely across the globe, causing a severe humanitarian and economic burden, in addition to the burden on healthcare services. As of 12th February 2021, more than 107.4 million COVID-19 cases, and 2.4 million COVID-19 attributable deaths, were reported globally [4]. In Bangladesh, 539,571 COVID-19 positive cases and 8248 deaths due to COVID-19 were reported [4]. However, the actual number of cases and COVID-19-related deaths might be much higher in Bangladesh because of the inadequate testing and poor surveillance system [5].

Although many drugs are being tested, there is no approved treatment for COVID-19 infection [6]. The lack of effective treatment, the severe burden of COVID-19 infection, and its fast spread highlight the need for rapid vaccine development, leading to unprecedented scale and speed in the efforts to develop an efficacious and safe vaccine. This process was record-breaking, entering human clinical trial testing in March 2020 [7], and by April 2020, more than 100 vaccine candidates were being developed using various methods, five of which were in the clinical evaluation stage and 73 at exploratory or pre-clinical stage [7,8]. As of 12 November 2020, the number of candidate vaccines in preclinical evaluation has grown to 164. Among the 48 vaccines which were in the clinical evaluation stage, four vaccines so far have been cleared for phase three trials: Pfizer/BioNTech’s BNT162b2, Moderna’s mRNA-1273, University of Oxford and AstraZeneca’s AZD1222, and Gamaleya’s Sputnik V vaccine [9]. Among these, Pfizer/BioNTech’s vaccine and two versions of the AstraZeneca/Oxford vaccine against SARS-CoV-2 infection produced by AstraZeneca-SKBio (Republic of Korea) and the Serum Institute of India are listed by WHO for emergency use [10].

Vaccine hesitancy is one of the ten most serious threats to global health. As reported by WHO (2019), reasons behind refusal or unwillingness can be related to inconvenience in accessing vaccines, complacency, or lack of trust [11]. However, there are a variety of unclassifiable factors associated with vaccine hesitancy with different, unmeasurable levels of influence, rendering vaccine hesitancy a complex matter that varies depending on the context, time, place, and vaccine [12]. Vaccine hesitancy affects population coverage and thus affects the containment of the targeted diseases. A study conducted in the USA [13] shows that with 80% vaccine efficacy and 10% mask usage, COVID-19 may be eliminated from the USA with at least 90% population coverage. The population coverage percentage can be decreased to 82% if mask compliance increases to 50% of the population.

There are differences in the acceptance rate of vaccination against SARS-CoV-2 across a sample population of 19 different countries: from less than 55% in Russia to 90% in China [14]. In a global overview of 20 studies, vaccine hesitancy for the not yet available vaccine differed from very low in China to very high (more than 40%) in Czechia Turkey [15]. A positive attitude toward vaccination against SARS-CoV-2 was associated with a high level of education, income, medium and increased numbers of cases and fatality rates, and trust in the government [14].

In the first half of 2020, the intent to receive a vaccine against SARS-CoV-2, if it became available, was different in different countries. For example, it was 64.7% in Saudi Arabia [16], >72% in China [17], and only 20% of participants surveyed in the USA expressed unwillingness to take the vaccine [18]. In England, the majority expressed wiliness to accept a vaccine against SARS-CoV-2 for themselves—55.8% said definitely; 34% were unsure but leaning towards yes [19]. However, the mentioned studies’ data on intent to receive hypothetical vaccine were mainly collected in the first half of 2020, which raises the question of whether the individuals who expressed their willingness to take the vaccine are still willing to take it after it is made available to the public, and after much time has passed. According to a study conducted in the USA among more than 7000 individuals, willingness to receive a vaccine against SARS-CoV-2 declined from 71% in April to 53.6% in October 2020. In contrast, the percentage of unwillingness to take the vaccine increased from 18% to 32%. Most of the individuals who were unwilling to be vaccinated feared the side effects and long-lasting health complications, in addition to being uncertain of the vaccine’s effectiveness [20].

With inept and inadequate healthcare systems, Bangladesh has been attempting to contain the impact of the COVID-19 pandemic since March 2020 [21]. Bangladesh severely lacked the preparedness to tackle the spread of COVID-19 [22], and the country’s COVID-19 testing system has also been criticized at a global level [5]. However, Bangladesh has experience of running successful immunization programs for children. The success of any vaccination program depends on widespread public education campaigns regarding vaccine safety and efficacy. It is time to see whether the country is ready to accept new vaccines against SARS-CoV-2. The Bangladesh government launched the SARS-CoV-2 vaccination program on 27 January 2021 and announced that interested individuals could register their interest to get vaccinated through a designated website. This process is not convenient for the elderly and rural population groups [23]. A recent global survey on vaccine confidence and barriers to vaccine uptake revealed that the highest estimated percentage of respondents from Bangladesh strongly agreed that vaccines are necessary and safe to use to control an infectious disease [24]. However, the lower official mortality rate due to COVID-19 infection and the lower official rate of infection among the population may also work against vaccine uptake [23].

The COVID-19 pandemic profoundly increased fiscal imbalances in many low- and middle-income countries. An estimated 80% of individuals in these countries will not benefit from the COVAX-provided vaccine [25]. Therefore, many low- and middle-income countries may not be able to provide free vaccines to the entirety of their population. This makes it important to assess the willingness to pay (WTP), an approach to estimate the maximum amount that an individual is willing to spend on a health service for increased health benefits [26].

It is crucial to study the current level of acceptance and WTP for vaccination against SARS-CoV-2 to assist the government and public health officials in addressing vaccine hesitancy, equitable access, and pricing of the vaccine, and to develop proper educational materials to build vaccine literacy encouraging the public to receive safe and efficient immunization. In this context, this article reports the intent to receive the vaccine against SARS-CoV-2 offered by the Government of Bangladesh, the sociodemographic predictors of intent, and the health belief model (HBM) constructs that can predict the intent to receive the vaccine in Bangladesh. In addition, this article reports the WTP for a vaccine against SARS-CoV-2 and vaccine preferences among the population of Bangladesh.

## 2. Materials and Methods

We conducted an online cross-sectional survey from 20th January 2021 to 27th January 2021 in Bangladesh. To disseminate our online questionnaire, we used all commonly used social media platforms in Bangladesh, such as Facebook, WhatsApp, and LinkedIn, and emails were sent to professional networks to disseminate the questionnaire. Every individual completing our questionnaire was requested to forward the survey link to their network. Participants were informed about our questionnaire dissemination platforms and were requested to avoid multi-registration. By the end of the data collection period, 697 individuals responded to our invitation. Any individuals living in Bangladesh and aged 18 years or over were considered eligible for this study.

Our questionnaire included questions on sociodemographic variables, health status, individual and family history of SARS-CoV-2 infection, intention to receive a vaccine against SARS-CoV-2, and HBM constructs related to vaccination against SARS-CoV-2. Participants’ intention to receive a vaccine against SARS-CoV-2 was assessed using a one-item question (‘If a vaccine against COVID-19 infection is available for you, would you take it?’) on a four-point Likert type scale (‘No, definitely not’ to ‘Yes, definitely’). We developed the questionnaire by reviewing the relevant literature. To ensure the relevance and clarity of the questions, a panel of public health scientists assessed the content validity of the questionnaire. Moreover, this questionnaire was pilot tested among a group of university students.

To assess the HBM constructs, we asked three questions to determine perceived susceptibility to COVID-19 infection, three questions to assess perceived severity of COVID-19, two questions to assess the perceived benefits of a vaccine against SARS-CoV-2, five questions to assess perceived barriers to getting a vaccination against SARS-CoV-2, and two questions to assess cues to action. We used simplified response options—agree/disagree—since we conducted an online, self-administered survey.

We assessed willingness to pay for a vaccine against SARS-CoV-2 using two questions on whether the participants were willing to pay for a vaccine and the maximum amount that they were willing to pay in Bangladeshi taka (BDT). In reporting, we converted the local currency, BDT, into United States Dollar (USD). One USD is equivalent to 84.8 BDT (https://www.bb.org.bd/econdata/exchangerate.php (accessed on 23 February 2021). In addition, we assessed the participants’ vaccine preferences, such as whether they wished to receive a vaccine that would be available in private sectors instead of the free vaccine provided by the government, and whether the country of vaccine origin mattered to them.

We first generated frequency tables for all the variables included in the questionnaire; we examined the tables for discrepancies, incorrect reporting, outliers, and patterns. Descriptive analyses were performed to report the distribution of sociodemographic, health status, vaccination intent variables, and HBM constructs among the participants, willingness to pay for a vaccine, and vaccine preferences. The amount that the participants were willing to pay was not normally distributed. Therefore, we reported the median amount that the participants were willing to pay for a vaccine against SARS-CoV-2. Finally, we performed logistic regression analyses to identify the most relevant and significant determinants of vaccination intent against SARS-CoV-2 in terms of sociodemographic characteristics and HBM constructs. We reported the odds ratio (OR) with 95% confidence interval (CI), depicting the likelihood of vaccine acceptance by variables from the HBM and the sociodemographic variables from the logistic regression models. In multivariable logistic regression analyses, hypothesized sociodemographic predictors of SARS-CoV-2 vaccination intent investigated included gender, age, education, occupation, area of residence, religion, history of chronic disease, and SARS-CoV-2 infection in participants and family. While the hypothesized HBM constructs investigated included perceived susceptibility to SARS-CoV-2 infection, severity of COVID-19 illness, benefits of a SARS-CoV-2 vaccine, and barriers to getting a vaccination against SARS-CoV-2 and cues to action. We performed three multivariable logistic regression analyses to investigate the predictors of vaccination intent: participants with a definite intent versus other intent groups, participants with a definite or probable intent versus other intent groups, and participants with a definite intent against vaccination versus other intent groups. All the statistical analyses were performed using IBM SPSS Statistics 20.

Ethical approval was obtained from the Research Ethics Committee of Sapporo Dental College and Hospital, Dhaka, Bangladesh on 20 January 2021 (reference number: SDC/C-7/2021/784). All participants were informed about the objectives of the study. They were also informed that participating in this study was entirely voluntary, and participation and non-participation were not associated with any benefit or harm. The first page of the survey form contained an informed consent form.

## 3. Results

### 3.1. Participants’ Characteristics

A total of 697 participants completed the online questionnaire. Table 1 presents their sociodemographic and health-related background. Our sample had fair representation of both genders, with 53.4% male and 46.6% female participants, but was dominated by young adults aged 18–29 years (65.6%); residents of Dhaka (71.7%); individuals with tertiary-level education (81.8%); Muslims (94.4%); students (42.6%), and health professionals/workers (19.4%). Regarding health status, over one fifth of participants (20.5%) reported having a known diagnosis of any chronic disease, the majority (62.6%) reported having at least one family member with a known diagnosis of any chronic disease, 14.6% had contracted COVID-19, and over one quarter (28.1%) reported having at least one family member who had contracted COVID-19.

### 3.2. Perceived COVID-19-Related Health Beliefs

Figure 1 presents the perceived health beliefs of the participants in relation to the vaccine against SARS-CoV-2. Most of the participants believed that they were susceptible to COVID-19 infection, 59.3% felt that, in their current situation, contracting COVID-19 was very likely, and 60.3% felt that the possibility of contracting COVID-19 in the future was very high for them. Most of them also perceived COVID-19 as a severe disease, with 84.2% considering COVID-19 complications as very serious and even life-threatening; 60.5% were afraid that they would become very sick if they contracted COVID-19, and 56.4% were afraid of COVID-19. Most of the participants considered the SARS-CoV-2 vaccine beneficial, with 74.3% reporting that the vaccine would make them feel less worried about contracting COVID-19, and 73.5% believed that vaccination against SARS-CoV-2 would decrease their chances of contracting COVID-19. However, 77.5% of the participants were worried that side effects of the vaccine would interfere with their usual activities, 81.8% doubted the efficacy of the vaccine, and 83.1% were concerned about the vaccine’s safety. Among the perceived barriers, most participants were not concerned about affordability and the vaccine being ‘halal’ or not, with only 46.9% being concerned about affordability, and 34.7% were concerned with the vaccine being ‘halal’ or not. Regarding cues to action, 91.1% reported that they would take the vaccine only after receiving complete information on its composition, safety, and efficacy, and 67.1% claimed that they would take the vaccine after many others had already received it.

### 3.3. COVID-19 Vaccination Intent and Predictors

Figure 2 presents various degrees of vaccination intention against SARS-CoV-2 among the participants. Most of the participants demonstrated some degree of intent, with 26% reporting that they would definitely receive the vaccine and 43% stating that they would probably receive the vaccine. However, 7% demonstrated a definite negative intention and the remaining 24% indicated a probable negative intention.

Table 2 presents the sociodemographic predictors of SARS-CoV-2 vaccination intent. We found no evidence of a significant association between definite SARS-CoV-2 vaccination intent and sociodemographic variables—gender, age, education, occupation, area of residence, religion, and health status. However, a significant association was observed between definite vaccination intent and participants’ previous COVID-19 infection status, with the odds of definite vaccination intent among the individuals with previous COVID-19 infection being 2.86 (95% CI: 1.71–4.78) times the odds among individuals without any history of the disease, adjusting for the effect of all other sociodemographic variables.

In addition to a definite or probable intent of receiving vaccination against SARS-CoV-2, we found evidence of a signification association with occupation and religion. Individuals in the private sector were 60% less likely to have a definite or probable vaccination intent than unemployed individuals/housewives (OR: 0.40; 95% CI: 0.18–0.90). However, no significant association was found with other occupational groups. We also found that Muslims were 64% less likely to have a definite or probable vaccination intent than non-Muslims given that the effects of other variables are kept constant (OR: 0.36; 95% CI: 0.15–0.89).

Regarding definite negative intent, we found a statistically significant association with the area of residence after adjusting for the effect of other sociodemographic variables. Individuals living outside of Dhaka were 70% less likely to have a definite negative intent of vaccination against SARS-CoV-2 than those living in Dhaka (OR: 0.30, 95% CI: 0.13–0.74).

Table 3 shows an association between HBM constructs and different degrees of vaccination intent against SARS-CoV-2. We found no evidence of association between perceived susceptibility constructs and definite vaccination intent. Our findings suggest a statistically significant association between definite vaccination intent and the perception that complications from COVID-19 are serious. Individuals who perceive that complications from COVID-19 are serious are more likely to have a definite intent (OR: 1.93, 95% CI: 1.04–3.59). In addition, our results suggest that individuals with the belief that vaccination makes them feel less worried about catching COVID-19 (OR: 4.42, 95% CI: 2.25–8.68), and individuals who are concerned about the affordability of the vaccine (OR: 1.51, 95% CI: 1.01–2.25), have significantly higher odds of having a definite vaccination intent than individuals without such belief and concern. On the other hand, individuals who are concerned that possible side effects of the vaccine might interfere with their usual activities (OR: 0.34, 95% CI: 0.21–0.53) and individuals who would take the vaccine if the vaccine was taken by many others (OR: 0.44, 95% CI: 0.29–0.67) were less likely to have a definite intent than the other groups.

Regarding an association between definite or probable vaccination intent against SARS-CoV-2, our results suggest that individuals who believe that vaccination will decrease their chance of getting COVID-19 are 2.23 times more likely to have vaccination intent compared to individuals who do not believe so (OR: 2.23, 95% CI: 1.39–3.59), adjusting for the effect of all other HBM items. In addition, individuals who think that vaccination will make them feel less worried about contracting COVID-19 have higher odds of definite or probable vaccination intent (OR: 4.31, 95% CI: 2.73–6.82) than individuals who do not think so, adjusting for the effect of all other HBM construct items. On the other hand, individuals who are concerned about the side effects of the vaccine against SARS-CoV-2 are 51% less likely to have vaccination intent compared to individuals who are not concerned with vaccine-related side effects (OR: 0.49, 95% CI: 0.29–0.84), adjusting for the impact of other HBM construct items.

Regarding the association between definite negative vaccination intent against SARS-CoV-2 and HBM construct items, our results suggest evidence of a significant association between definite negative intent and the concern that the vaccine may not be halal and the belief that vaccination will make them less worried about contracting COVID-19. Individuals concerned that the vaccine against SARS-CoV-2 may not be halal are 2.03 times likely to have a definite negative intent than the individuals who do not have such a concern (OR: 2.03, 95% CI: 1.04–3.96), adjusting for the effect of other HBM items. On the other hand, individuals who think that vaccination will make them feel less worried about contracting COVID-19 are 80% less likely to have a negative intent compared to the individuals without such a feeling (OR: 0.20, 95% CI: 0.09–0.46), adjusting for the effect of other HBM items.

### 3.4. Willingness to Pay and Vaccination Preference

Table 4 presents the willingness to pay and other preferences for vaccination against SARS-CoV-2. Our results suggest that most people (68.4%) are willing to pay for the COVID-19 vaccine. The median amount that they are willing to pay is USD 7.08. Our results also suggest that for over half of the participants (52.2%), the country of origin of the vaccine is important. Furthermore, one third (32.9%) of the population stated that they would prefer to purchase a vaccine from the private sector’s available alternatives than receive the free vaccine offered by the government.

## 4. Discussion

This article reports intent to receive a vaccine against SARS-CoV-2 in Bangladesh, and how the health belief model (HBM) constructs help to predict this intent. The study findings have shown that only 7% of the participants definitely and 24% probably will not accept the vaccine, while 26% participants definitely and 43% probably will receive the vaccine. Our study suggests a lack of association between the different degrees of vaccination intent and most sociodemographic variables but a significant association with most HBM constructs.

Although this study found no evidence of an association between SARS-CoV-2 vaccination intent or hesitancy and most sociodemographic variables (age, gender, education), studies conducted elsewhere reported otherwise. A recent study in Ireland and the UK revealed that younger (18–24 years) individuals are more vaccine-hesitant [27]. In the USA, an estimated 86% of people over 61 years of age are more likely to receive the vaccine against SARS-CoV-2 [28]. Another study in Jordan and Kuwait found that the SARS-CoV-2 vaccine acceptance rate is higher among males and people with higher educational attainment [29].

Surprisingly, our study suggests that, compared to Dhaka city residents, residents of other cities are significantly less likely to have a definite intent against vaccination. This is probably because the SARS-CoV-2 -vaccination awareness programs are not running adequately. Many people are inclined to believe rumors that are not based on science because of their reliance on social media for information. Furthermore, this could also be attributable to people’s mistrust of the government. Studies suggest that misinformation regarding side effects, rumors, and conspiracy theories about COVID-19 negatively affect the willingness of the population to be vaccinated [30,31,32,33]. This study also found that religious belief is an essential predictor of vaccine acceptance. A longitudinal survey in Australia found that people with higher religiosity levels are more likely to be hesitant [34]. Our study also suggests that people previously infected with COVID-19 are almost three times more likely to accept the vaccine against SARS-CoV-2 than the uninfected population. This is in line with other study findings in Saudi Arabia [16] and France [35]. COVID-19 negatively affects not only one’s physical health but also one’s psycho-social wellbeing. Hence, their negative experience of suffering from COVID-19 is perhaps the reason for their higher intent to receive the vaccine.

The health belief model analysis revealed that perceived severity, perceived benefits, and barriers are associated with the intention to receive the vaccine. Individuals who consider the complications of COVID-19 to be very serious are more likely to be vaccinated. In addition, those who believe that vaccination will decrease their chance of contracting COVID-19 express definite or probable intent to be vaccinated. Those who have experienced COVID-19 accept the fact that the vaccine can interrupt the transmission of COVID-19. It should be noted that, in order to investigate HBM predictors of vaccination intent, we did not stratify participants by their history of SARS-CoV-2 infection. Although a history of SARS-CoV-2 infection may influence HBM, investigating such an association was not the focus of this paper.

Some individuals are concerned that the vaccine may not be halal, and they are more likely to be unvaccinated. Halal may be defined as something that is permitted according to Islamic law. Typically, halal is the permissibility to eat, drink, or do something based on Islamic law and principles. A recent study conducted in Malaysia on vaccine development showed their willingness towards trusting ‘halal’ vaccines. Malay Muslim children’s parents have expressed concerns regarding the vaccine ingredients’ halal state because they believe that imported vaccines might be composed of porcine-derived agents, such as deoxyribonucleic acid (DNA), which is not halal, as the utilization of porcine-sourced products, including medicines, is generally not permissible for Muslims [36]. For vaccination against SARS-CoV-2, similar concerns arose in Indonesia [37]. Generally, people are receiving mixed information about vaccines against SARS-CoV-2. In particular, social media has played a large role in spreading a lot of antivaccination misinformation and rumors that vaccines are not halal. Public health-related misinformation spread both in social platforms and digital media has become vital for the ongoing second wave, leading to panic regarding SARS-CoV-2 vaccine updates [38]. From the Islamic point of view, preservation of life is secondary to the preservation of religion, and Muslims who refuse to receive vaccines on the grounds that they are non-halal is a significant problem in many Muslim countries [39].

Our results suggest that the vast majority of people (68.4%) are willing to pay for the vaccine and the average amount that they are ready to pay for the vaccine is USD 7.08. Studies conducted in Indonesia and Ecuador found that an estimated 78.3% and 85% of the population are prepared to pay for the vaccine, respectively, and individuals from Australia are willing to pay to reduce the waiting time limit [40,41,42]. Although the majority of people are willing to pay for a vaccine, we found that individuals concerned about vaccine affordability are more likely to have definite vaccination intent. It is possible that those do not have a definite intent do not care about vaccine affordability, but this is unlikely considering that the Bangladesh government may not be able to cover the huge population or might take years to cover the population. Therefore, participants with a definite intent might have to resort to private providers if the vaccines are available from them, and this is where the concerns regarding affordability arise. However, given the close results, we should be cautious in stating that concerns about affordability predict vaccine intent. Further studies are recommended to investigate this association. The study participants also shared that vaccine origin is an important criterion for them to accept the vaccine and they preferred vaccines developed and manufactured overseas. A contrasting result reported from participants in China was a preference for a locally manufactured over a foreign-made vaccine against SARS-CoV-2 [43].

Bangladesh is hosting and sheltering over a million Rohingya refugees residing in Cox’s Bazar district, which is also a well-known tourist destination. The refugee camps are overcrowded, meaning that they are at risk of contracting SARS-CoV-2 [44]. These refugees have frequent interaction with the host communities, which puts the host community also at risk of SARS-CoV-2 infection.

The scientific community and Bangladesh government should work together to overcome this dilemma around vaccination against SARS-CoV-2 among the general population and use proper scientific evidence to educate the population. The current vaccination strategy implemented by the Bangladeshi health authorities should be followed and trusted by the general population. There should be adequate monitoring of social media platforms to stop the spread of misinformation and further research work is needed to understand common challenges to the acceptance of a new vaccine by the general population.

Although this research will make an apt contribution to the academic literature in this field, there are several limitations. Firstly, there are limited internet services in Bangladesh, so the online survey questionnaire was distributed to individuals with an internet connection. Moreover, the majority of the participants in the survey were educated and health-conscious Bangladeshi residents. Hence, t might affect the generalizability of the study. Another limitation is the use of a convenient sampling method to recruit the participants; hence, the findings of the study cannot be generalized to the whole population.

## 5. Conclusions

The study findings have revealed that the intent to receive a vaccine against SARS-CoV-2 among the general population varies depending on their knowledge of COVID-19 transmission, and there was no association found with sociodemographic variables. Other important factors associated with vaccine hesitancy are concerns about the side effects and the question of whether the vaccine would be halal. However, individuals who believe that taking the vaccine will reduce their chance of contracting the disease are more likely to receive the vaccine.

## Figures and Tables

**Figure 1 vaccines-09-00416-f001:**
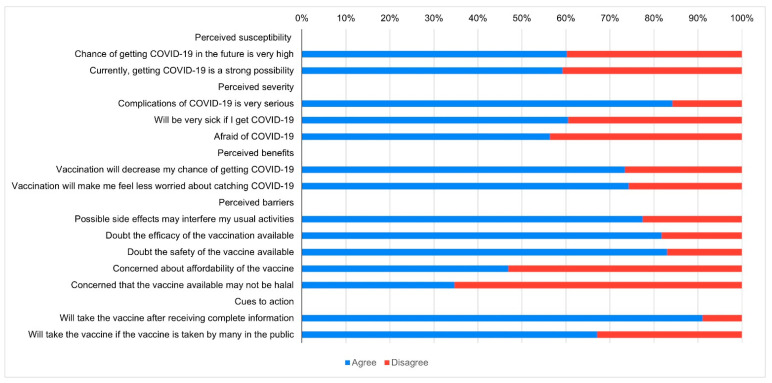
COVID-19-related health beliefs in Bangladesh.

**Figure 2 vaccines-09-00416-f002:**
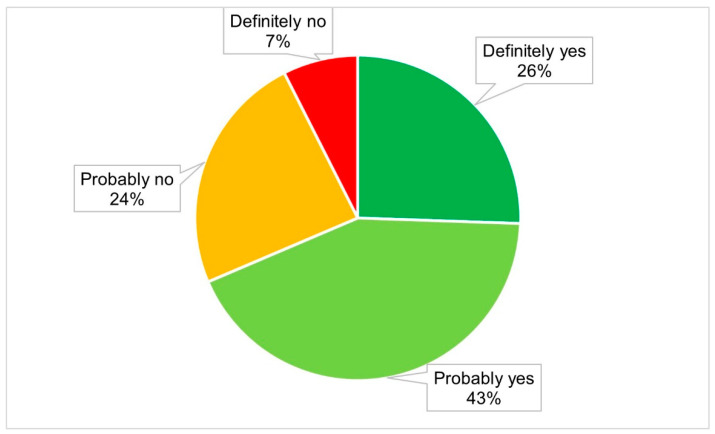
COVID-19 vaccination intent in Bangladesh.

**Table 1 vaccines-09-00416-t001:** Characteristics of the participants, COVID-19 vaccination intent and WTP, cross-sectional survey, January 2021, Bangladesh.

Characteristics	*n* (%)
**Gender**	
Male	372 (53.4)
Female	325 (46.6)
**Age**	
18–29 years	457 (65.6)
30–39 years	126 (18.1)
40–49 years	72 (10.3)
50 years or more	42 (6.0)
**Education**	
Primary or below	19 (2.7)
Secondary	108 (15.5)
Tertiary (college/university)	570 (81.8)
**Occupation**	
Unemployed	58 (8.3)
General worker	24 (3.4)
Student	297 (42.6)
Self-employed	38 (5.5)
Private service	95 (13.6)
Government service	50 (7.2)
Health professionals	135 (19.4)
**Area of residence**	
Dhaka	500(71.7)
Outside of Dhaka	197(28.3)
**Religion**	
Islam	658 (94.4)
Other	39 (5.6)
**Participant has chronic disease**	
No	554 (79.5)
Yes	143 (20.5)
**Family member has chronic disease**	
No	261 (37.4)
Yes	436 (62.6)
**Participant diagnosed with COVID-19**	
No	595(85.4)
Yes	102(14.6)
**Family member diagnosed with COVID-19**	
No	501 (71.9)
Yes	196 (28.1)

**Table 2 vaccines-09-00416-t002:** Sociodemographic predictors of SARS-CoV-2 vaccination intent in Bangladesh, multivariable logistic regression analyses.

Characteristics	Definite Intent	Definite or Probable Intent	Definite Intent against Vaccination
	OR (95% CI)	OR (95% CI)	OR (95% CI)
**Gender**			
Female	1	1	1
Male	1.32 (0.91–1.92)	1.20 (0.84–1.70)	0.63 (0.32–1.21)
**Age**			
18–29 years	1	1	1
30–39 years	1.11 (0.63–1.99)	0.87 (0.52–1.45)	0.80 (0.29–2.24)
40–49 years	1.37 (0.67–2.80)	1.04 (0.53–2.01)	2.60 (0.96–7.07)
50 years or more	1.78 (0.70–4.53)	0.86 (0.37–1.98)	3 (0.87–10.24)
**Education**			
Tertiary (college/university)	1	1	1
Secondary	0.90 (0.53–1.52)	0.99 (0.61–1.60)	2.01 (0.91–4.45)
Primary or below	0.99 (0.24–4.05)	0.67 (0.22–1.98)	2.81 (0.64–12.36)
**Occupation**			
Unemployed	1	1	1
General worker	0.53 (0.12–2.35)	0.56 (0.18–1.69)	4.30 (0.93–19.84)
Student	1.71 (0.73–4.03)	0.59 (0.26–1.30)	1.04 (0.28–3.84)
Self-employed	0.71 (0.22–2.24)	0.48 (0.18–1.24)	3.33 (0.81–13.65)
Private service	0.58 (0.22–1.52)	0.40 (0.18–0.90) *	1.97 (0.52–7.40)
Government service	1.82 (0.70–4.70)	0.70 (0.27–1.74)	2.96 (0.74–11.40)
Health professionals	2.06 (0.88–4.83)	0.71 (0.32–1.59)	0.73 (0.18–3.01)
**Area of residence**			
Dhaka	1	1	1
Outside of Dhaka	0.77 (0.51–1.16)	0.85 (0.58–1.22)	0.30 (0.13–0.74) **
**Religion**			
Other	1	1	1
Islam	0.52 (0.25–1.08)	0.36 (0.15–0.89) *	1.11 (0.24–5.17)
**Participant has chronic disease**			
No	1	1	1
Yes	0.68 (0.40–1.15)	1.05 (0.67–1.67)	0.44 (0.18–1.07)
**Family member with chronic disease**			
No	1	1	1
Yes	0.95 (0.65–1.40)	1.23 (0.87–1.75)	1.07 (0.57–2.04)
**Participant’s history of SARS-CoV-2 infection**			
No	1	1	1
Yes	2.86 (1.71–4.79) **	1.25 (0.74–2.11)	0.52 (0.16–1.65)
**Family history of SARS-CoV-2 infection**			
No	1	1	1
Yes	0.74 (0.48–1.17)	1.05 (0.70–1.59)	1.23 (0.59–2.54)

* Statistically significant at *p* < 0.05; ** statistically significant at *p* < 0.01.

**Table 3 vaccines-09-00416-t003:** Health belief model constructs and SARS-CoV-2 vaccination intent in Bangladesh, multivariable logistic regression analyses.

HBM Constructs	Definite Intent	Definite or Probable Intention	Definite Intent against Vaccination
	OR (95% CI)	OR (95% CI)	OR (95% CI)
**Perceived susceptibility**			
Chance of getting COVID-19 in the future is very high	0.70 (0.43–1.14)	0.53 (0.33–0.83) **	2.01 (0.93–4.36)
Currently, getting COVID-19 is a strong possibility	0.88 (0.54–1.42)	0.80 (0.50–1.27)	0.97 (0.44–2.14)
**Perceived severity**			
Complications of COVID-19 is very serious	1.93 (1.04–3.59) *	0.91 (0.53–1.60)	0.83 (0.34–2.02)
Will be very sick if I get COVID-19	0.81 (0.52–1.26)	0.79 (0.50–1.22)	1.28 (0.56–2.89)
Afraid of COVID-19	1.17 (.72–1.74)	0.99 (0.64–1.52)	1.10 (0.51–2.38)
**Perceived benefits**			
Vaccination will decrease my chance of getting COVID-19	1.71 (0.94–3.14)	2.23(1.39–3.59) **	0.59(0.26–1.33)
Vaccination will make me feel less worried about catching COVID-19	4.42 (2.25–8.68) **	4.31 (2.73–6.82) **	0.20 (0.09–0.46) **
**Perceived barriers**			
Possible side effects may interfere my usual activities	0.34 (0.21–0.53) **	0.49 (0.29–0.84) **	2.53 (0.85–7.45)
Doubt the efficacy of the vaccination available	1.39 (0.72–2.67)	1.68 (0.94–3.02)	0.59 (0.23–1.49)
Doubt the safety of the vaccine available	0.62 (0.33–1.18)	0.79 (0.42–1.48)	1.29 (0.46–3.62)
Concerned about affordability of the vaccine	1.51 (1.01–2.25) *	1.30 (0.88–1.90)	1.40 (0.72–2.74)
Concerned that the vaccine available may not be halal	0.66 (0.42–1.01)	0.92 (0.62–1.38)	2.03 (1.04–3.96) *
**Cues to action**			
Will take the vaccine after receiving complete information	0.85 (0.41–1.75)	1.20 (0.61–2.34)	0.41 (0.16–1.04)
Will take the vaccine if the vaccine is taken by many in the public	0.44 (0.29–0.67) **	1.21 (0.80–1.83)	0.40 (0.20–0.79) **

* Statistically significant at *p* < 0.05; ** statistically significant at *p* < 0.01.

**Table 4 vaccines-09-00416-t004:** Willingness to pay and preference for vaccine against SARS-CoV-2 in Bangladesh.

Willingness to Pay and Preference	Participants
Willing to pay	68.4%
Median amount willing to pay in USD	7.08
Prefer buying a vaccine from available alternatives over the free one provided by the government	32.9%
Country of origin of the vaccine matters	52.2%

## Data Availability

The data presented in this study are available on reasonable request from the corresponding author.

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
