# Peer review of "COVID-19 Vaccination Intent and Willingness to Pay in Bangladesh: A Cross-Sectional Study"

_vaccines, 2021, doi:10.3390/vaccines9050416_

Round 1

Reviewer 1 Report

In this manuscript, Ilias Mahmud et al. assessed the COVID-19 vaccination intention and willingness to pay using questionnaire.

Although the study participants do not fully represent the general population in Bangladesh, as the authors noted, most of the results are in line with what’s known in the field, for example the experience of SARS-CoV-2 infection and the belief that complications of COVID-19 is serious related strongly with definite intent in vaccination, etc. However, the finding that concerns about vaccine affordability significantly related with the definite intent in vaccination is somehow counterintuitive. In fact, the OR and 95% CI of definite intent and definite intent against vaccination are very close. More information or discussion is needed to help readers better understand this finding.

In Material and Methods section, the authors mentioned various social media groups and email networks were used to disseminate the questionnaire. Given this, one may receive the study invitation multiple times. Therefore, it’s important to make sure there’s no multi-registration within the 697 questionnaires. The authors are invited to elaborate on the possibility of this multi-registration issue and how its was excluded from data analyses if applicable.

Also in Material and Methods section, the authors described in detail the questions and options of answers in the questionnaire. But how the statistical analyses were performed especially the model(s) used in multi-variable analyses such as logistic regression is not fully provided. If possible, a sample of the questionnaire as supplementary material and more in-depth information on data analyses could be useful for readers to better understand the study.

Minor comments:

  1. Section 3.3 first paragraph, the percentage of definitely yes is described as 25.5% and rounded up to 26% in figure 2; while the percentage of definitely no is written as 7.5% and rounded down to 7% in the figure. It might be better to keep the two sets of numbers in text and figure consistent.
  2. Some of the text could be more concise, for example “adjusting for the impact of other HBM construct items” appears more than five times in section 3.3.

Author Response

Although the study participants do not fully represent the general population in Bangladesh, as the authors noted, most of the results are in line with what’s known in the field, for example the experience of SARS-CoV-2 infection and the belief that complications of COVID-19 is serious related strongly with definite intent in vaccination, etc. However, the finding that concerns about vaccine affordability significantly related with the definite intent in vaccination is somehow counterintuitive. In fact, the OR and 95% CI of definite intent and definite intent against vaccination are very close. More information or discussion is needed to help readers better understand this finding.

We classified our participants into three groups based on their vaccination intent: definite intent, definite or probable intent and definite intent against vaccination.

Regarding the association between the concerns of vaccine affordability and definite intent: participants with such concerns are more likely to have a definite intent. Perhaps those do not have a definite intent does not care about vaccine affordability but those have definite intent they care about vaccine affordability considering the huge population in Bangladesh and since government may not be able to cover the population or might take years to cover the population. Therefore, participants with a definite intent might have to resort to private providers if the vaccines are available from them and here comes the concerns of affordability

The odds of definite intent and definite intent against vaccination are not same. Regarding the association between intent and socio-demographic variables, significant association was observed between definite intent and history of SARS-CoV-2 infection [2.86, 95% CI: 1.71 – 4.79]. While for the definite negative intent significant association was with the area of residence [OR: .30, 95% CI: .13 – .74]. Please see Table 2. While, regarding the association between the HBM constructs and intent, we found significan association between ‘Vaccination will make me feel less worried about catching COVID-19’ and both definite intent and definite negative intent but if you look at the odds ratio you will see that the association is in the opposite direction. Participants with a believe that Vaccination will make me feel less worried about catching COVID-19 are more likely [4.42, 95% CI: 2.25 – 8.68] to have a definite intent but less likely [OR: .20, 95% CI: .09 – .46] to have a definite negative intent. Similar patterns of association were also observed with other HBM constructs. Please see Table 3 (first and third column).   In Material and Methods section, the authors mentioned various social media groups and email networks were used to disseminate the questionnaire. Given this, one may receive the study invitation multiple times. Therefore, it’s important to make sure there’s no multi-registration within the 697 questionnaires. The authors are invited to elaborate on the possibility of this multi-registration issue and how its was excluded from data analyses if applicable.     We agree, one may have received the invitation from more than one source. Participants were requested to complete only time. We believe our unique template and short duration of data collection (one week) helped the participants to avoid multi-registration. We have clarified this in methods section. Please see first paragraph of methods section. Changes are highlighted in red.   Also in Material and Methods section, the authors described in detail the questions and options of answers in the questionnaire. But how the statistical analyses were performed especially the model(s) used in multi-variable analyses such as logistic regression is not fully provided. If possible, a sample of the questionnaire as supplementary material and more in-depth information on data analyses could be useful for readers to better understand the study.   We have provided the questionnaire as a supplementary material and more details on analyses performed are added. Please see page number 4 paragraph number 2.   Section 3.3 first paragraph, the percentage of definitely yes is described as 25.5% and rounded up to 26% in figure 2; while the percentage of definitely no is written as 7.5% and rounded down to 7% in the figure. It might be better to keep the two sets of numbers in text and figure consistent.   Thanks! We have corrected these inconsistencies. Correct statistics are 26% and 7%, respectively. Please see page 6 paragraph 1.   Some of the text could be more concise, for example “adjusting for the impact of other HBM construct items” appears more than five times in section 3.3.   We agree. Since these are multivariable logistic regression analysis output, it implies that it was adjusted for the other varaibles included in the model. Hence, we have deleted “adjusting for the impact of other HBM construct items”. See the manuscript.  

Reviewer 2 Report

The paper addresses a present and relevant topic, namely COVID-19 vaccine acceptance. Albeit methodologically weak (as an online survey was used and the study population is not representative), this piece of information could be still useful. Nevertheless, there are many major concerns that should be addressed that I have reported hereafter.

Abstract

  1. The abstract lacks reporting important information such as the setting and the study population of the study and its conclusions do not reflect the results as the paper does not address misinformation.

Introduction

  1. A time reference should be included for the following sentence “In Bangladesh, 539, 571 COVID-19 tested positive cases and 8248 COVID-19 caused deaths were reported [32].”
  2. From the paragraph on vaccine hesitancy onwards, several concepts were reported in a disconnected way. I would recommend the Authors drawing the attention on the importance of achieving high coverage (but without reporting data on efficacy because they are not useful in my view for demonstrating that the achievement of high coverage is important) and on the assessment of attitudes and behaviours with respect to COVID-19 vaccines. Actually, the most of evidence is on attitudes as vaccination campaigns were launched recently. Furthermore, the introduction does not report anything about the willingness to pay. Nonetheless, this aspect was also included in the title of the paper.  
  3. Eventually, the introduction should be revised in order to make it more concise and focused on the problem.

Methods

  1. The survey was launched online. Indeed, it is not correct to report “By the end of January, when we closed our survey, 700 individuals received our invitation, of which three refused participation. Hence, our final sample size included 697 individual”. I guess that more individuals were reached by the invitation and that this number cannot be predicted. Please clarify this point.
  2. Methods do not report anything about some results reported in the text and the willingness to pay. Nonetheless, this last aspect was also mentioned in the title of the paper.  

Results

  1. The analysis on perceived COVID-19 related health beliefs should be restricted to specific groups (i.e. to people who did not have COVID for self-perceived risk). Putting together data returns a confused overview in my opinion.
  2. In my view the analysis of the three endpoints (Definite intent; Definite or probable intent; Definite intent against vaccination) is confusing. It was not anticipated in methods and it should be. Furthermore, I will limit the analysis to the evaluation of the vaccine positive attitudes maybe considering the first two endpoints if Authors want to make a sort of sensitivity analysis. Nevertheless, the results are different, and Authors should elaborate on these differences.
  3. I think that the analysis of Health belief model constructs and COVID-19 vaccination intent should be stratified according to previous COVID-19 infection.
  4. The analysis on WTP is disconnected from the rest of the paper. It is not described in methods and misses details (i.e. BDT meaning).

Discussion

  1. Discussion should be revised in the light of all previous comments.

Author Response

The abstract lacks reporting important information such as the setting and the study population of the study and its conclusions do not reflect the results as the paper does not address misinformation.   Done. See the manuscript   A time reference should be included for the following sentence “In Bangladesh, 539, 571 COVID-19 tested positive cases and 8248 COVID-19 caused deaths were reported [32].”   We used the reference manager EndNote. Unfortunately there was glitch in the output in the submitted manuscript version. We have updated every single references. Citations and their order are corrected. From the paragraph on vaccine hesitancy onwards, several concepts were reported in a disconnected way. I would recommend the Authors drawing the attention on the importance of achieving high coverage (but without reporting data on efficacy because they are not useful in my view for demonstrating that the achievement of high coverage is important) and on the assessment of attitudes and behaviours with respect to COVID-19 vaccines. Actually, the most of evidence is on attitudes as vaccination campaigns were launched recently. Furthermore, the introduction does not report anything about the willingness to pay.Nonetheless, this aspect was also included in the title of the paper.  

We have deleted some texts on efficacy. Following texts were deleted: A study in the USA [13] reported the need for at least 60% SARS-CoV-2 vaccine efficacy when 100% of the population is covered by vaccines. The efficacy rate increases to 80% with the dropping of the population coverage to 75%. However, for only 60% population coverage the vaccine must be 100% efficacious for the epidemic to be extinguished. Inadequate coverage will only reduce the peak of epidemic even with a vaccine with 100% efficacy.

We have added a paragraph on willingness to pay in the introduction.   Eventually, the introduction should be revised in order to make it more concise and focused on the problem.   Done. See the manuscript.   The survey was launched online. Indeed, it is not correct to report “By the end of January, when we closed our survey, 700 individuals received our invitation, of which three refused participation. Hence, our final sample size included 697 individual”. I guess that more individuals were reached by the invitation and that this number cannot be predicted. Please clarify this point.   We agree and revised this section accordingly. Current texts: By the end of the data collection period, 700 individuals responded to our invitation, of which three refused participation. Hence, our final sample size included 697 individuals.   Methods do not report anything about some results reported in the text and the willingness to pay. Nonetheless, this last aspect was also mentioned in the title of the paper.   Thanks for pointing this out! We have added information on williness to pay. Please see the revised methods highligted in red.   The analysis on perceived COVID-19 related health beliefs should be restricted to specific groups (i.e. to people who did not have COVID for self-perceived risk). Putting together data returns a confused overview in my opinion.   We did multivariable logistic regression. Which compared vaccination intent between gropus, adjusting for the effects of other variables included in the model.   In my view the analysis of the three endpoints (Definite intent; Definite or probable intent; Definite intent against vaccination) is confusing. It was not anticipated in methods and it should be. Furthermore, I will limit the analysis to the evaluation of the vaccine positive attitudes maybe considering the first two endpoints if Authors want to make a sort of sensitivity analysis. Nevertheless, the results are different, and Authors should elaborate on these differences.   We feel that it is important to understand the predictors of all these intent group. We have updated our methods section and informed the reader that we have run three multivariables logist regression analyssi to investigate the predictors of vaccination intent: definite intent vs other intent groups, definite or probable intent vs other intent groups, and definite intent against vaccination vs other intent group. Please see revised methods section highlighted in red.   I think that the analysis of Health belief model constructs and COVID-19 vaccination intent should be stratified according to previous COVID-19 infection.   Purpose of this analysis was to investigate the HBM predictors of vaccination intent. We understand that history of SARS-CoV-2 infection may influence the HBM constructs. But, that was not the focus of this study.   The analysis on WTP is disconnected from the rest of the paper. It is not described in methods and misses details (i.e. BDT meaning).

Thanks for pointing this out! We have added WTP in methods section and also in introduction section as you suggested earlier.

BDT is Bangladeshi currency. In revised mansucript we reported equivalent USD.   Discussion should be revised in the light of all previous comments.   Discussion is revised inlight of the revisions made in the manuscript based on review comments.      

Reviewer 3 Report

Vaccine hesitancy is one of the major factors influencing vaccine coverage and thus influencing the spreading of the target diseases. This manuscript reports the intent to receive a COVID-19 vaccine, its predictors and willingness to pay in Bangladesh. The research topic is interesting. However, the manuscript should be improved.

Following are the comments and suggestions:

  1. It was a cross-sectional survey. The total sample size is only 697 adults, which is too small for the whole country.
  2. It is an epidemiological study, but it doesn’t have statistical analysis.
  3. In addition to single-factor analysis, multi-factor analysis should be done so that important factors affecting vaccination intent immunization can be revealed.

Author Response

It was a cross-sectional survey. The total sample size is only 697 adults, which is too small for the whole country.   For a population survey with 104.87 million adult population (aged >18), 50% expected frequency (50% expected frequency gives the highest sample size), 5% acceptable margin of error (p<.05), 384 is the required sample size. However, we understand that a bigger sample size would be better.   It is an epidemiological study, but it doesn’t have statistical analysis.   We have performed statistical analysis (please see the methods and the results sections of the manuscript)   In addition to single-factor analysis, multi-factor analysis should be done so that important factors affecting vaccination intent immunization can be revealed. We have conducted multivariable (multi-factor) logistic regression. Please see the methods and results sections of the manuscript.    

Reviewer 4 Report

Dear editor, 

Dear authors,

I have read with an interest this paperThe authors took up a very current and important topic of COVID-19 vaccination intent and willingness to pay in Bangladesh. In my opinion, the manuscript in this shape requires revision. The following issues need correction:

Major:

  • The introduction should be more concise. Moreover, it should be concluded by the hypothesis of the study!
  • Only 700 people clicked on the provided link? Did you count every single clicked in link? If yes, the tool used to achieve this goal should be provided in the manuscript. If not, the sentence "(...) 700 individuals received our invitation, of which three refused participation" must be reconstructed.
  • Authors have stated that: "various social media groups and email network to disseminate the questionnaire. ". The full list of social media and email networks should be stated in the manuscript or in Supplementary materials.
  • The tables must be reconstructed. In this shape, they are "hard to read"...
  • Obtained results should be discussed with appropriate researches performed at the same time, e.g. https://www.mdpi.com/2076-393X/9/2/128 and https://www.mdpi.com/2076-393X/9/3/218
  • The strengths and limitation of the study should be stated in a separate section (after conclusions)

Minor

- Vaccines journal requires nonstructured abstract. Thus all headings (e.g. "Aim", "Subjects and methods",...) should be removed.

- References should be numbered: [1], [2], ... instead of [1], [22], [35], ... The authors may use e.g. Mendeley software to order their citation

- Some citations are provided as "(Lin et al, 2020)". It is not a style used in this journal.

- The authors stated that "We conducted an online cross-sectional survey in January 2021 in Bangladesh". Was it conducted 1-31 January? If not the proper days should be stated.

- The questionnaire used in the research with its English language version should be provided as Supplementary materials.

- The program used for statistical analysis must be provided (e.g. "Statistical analysis was performed using STATISTICA 13.1 (TIBCO, PaloAlto, Santa Clara, CA, USA).".

- Table 1: The data should be presented as n (%) instead of Per cent (count). Moreover, according to the MDPI policy, visible lines should be used to separate main groups of data (e.g. Gender, Age, ...), not every single row

- Please use the term vaccination against SARS-CoV-2 not COVID-19 vaccination

- the term "halal" should be explained

- Tables 2 and 3 are "hard to read" all numbers should be stated in one row (columns with numbers should be wider). The meaning of bold must be explained in Table's legend. 

- All figures and tables should be mentioned in the text of the manuscript (e.g. Table 2, Figure 2)

- Table 4: "Median amount willing to pay" should be provided in USD

Author Response

The introduction should be more concise. Moreover, it should be concluded by the hypothesis of the study!   We have further edited introduction section incorporating feedback. We did not write as hypothesis but provided objectives in the last paragraph of the introduction section. We believe this style is also acceptable.   Only 700 people clicked on the provided link? Did you count every single clicked in link? If yes, the tool used to achieve this goal should be provided in the manuscript. If not, the sentence "(...) 700 individuals received our invitation, of which three refused participation" must be reconstructed.   We have reconstructed this section. Please see the corresponding methods section highlighted in red.   Authors have stated that: "various social media groups and email network to disseminate the questionnaire. ". The full list of social media and email networks should be stated in the manuscript or in Supplementary materials.   Done. We have provided the list. Please see the manuscript   The tables must be reconstructed. In this shape, they are "hard to read"...   We have improved tables layout further. Hopefully, following journal editing it will be improved even further.   Obtained results should be discussed with appropriate researches performed at the same time, e.g. https://www.mdpi.com/2076-393X/9/2/128 and https://www.mdpi.com/2076-393X/9/3/218   We have cited these important references. Thanks for letting us know. Please see the manuscript.   The strengths and limitation of the study should be stated in a separate section (after conclusions)   We have followed the Vaccine journal submission format and kept the strengths and limitations of the study before the conclusion.   Vaccines journal requires nonstructured abstract. Thus all headings (e.g. "Aim", "Subjects and methods",...) should be removed.   Done― we have removed headings from abstract. Please see the manuscript.   References should be numbered: [1], [2], ... instead of [1], [22], [35], ... The authors may use e.g. Mendeley software to order their citation   Thanks for pointing this out. We used the reference manger EndNote. We have noticed a glitch in referencing in the submitted version. Not only these references but also all references citied. We have corrected all citations. It was not difficult since we had the original version with endnote field codes. Done. See the manuscript   Some citations are provided as "(Lin et al, 2020)". It is not a style used in this journal.   We have updated according to the journal format. Please see the revised manuscript.   The authors stated that "We conducted an online cross-sectional survey in January 2021 in Bangladesh". Was it conducted 1-31 January? If not the proper days should be stated.   We have provided dates in the revised manuscript   The questionnaire used in the research with its English language version should be provided as Supplementary materials.   We provided it as a supplementary material.   The program used for statistical analysis must be provided (e.g. "Statistical analysis was performed using STATISTICA 13.1 (TIBCO, PaloAlto, Santa Clara, CA, USA).".   We used SPSS version 20. It is now mentioned in the revised manuscript.   Table 1: The data should be presented as n (%) instead of Per cent (count). Moreover, according to the MDPI policy, visible lines should be used to separate main groups of data (e.g. Gender, Age, ...), not every single row   Done accordingly. Please see the revised table.   Please use the term vaccination against SARS-CoV-2 not COVID-19 vaccination   Done accordingly. Please see the manuscript   the term "halal" should be explained   Done accordingly.   Tables 2 and 3 are "hard to read" all numbers should be stated in one row (columns with numbers should be wider). The meaning of bold must be explained in Table's legend.    All numbers are arranged in one row. In bold we indicated statistically significant result. In revised tables, we indicated significant results with *, and indicated in table legend.   All figures and tables should be mentioned in the text of the manuscript (e.g. Table 2, Figure 2)   We cited all tables and figures in texts as you suggested.   Table 4: "Median amount willing to pay" should be provided in USD   We provided in USD in Table 4 and in texts  

Round 2

Reviewer 1 Report

The revision has addressed most of the comments regarding the first draft. However, there are two concerns remain unanswered:

  1. The statement "In fact, the OR and 95% CI of definite intent and definite intent against vaccination are very close." was referring to the correlations between vaccine intent and concerned about affordability of the vaccine. Sorry if it was not made clear enough in the first review report. In table 3, the OR (95% CI) numbers for definite intent, definite or probable intention, and definite intent against vaccination are 1.51 (1.01-2.25), 1.30 (0.88-1.90), and 1.40 (0.72-2.74), respectively. Authors did explain the relationship between the affordability and vaccine intent in the response. However, given these close results, one should be more careful in stating that concerned about affordability predicts vaccine intent.
  2. The percentage of definite yes and definite no appear in abstract, results text and figure 2. The discrepancy between results text and figure 2 were revised, but the numbers in abstract are still different from the others. Authors are invited to make the results consistent across the whole manuscript.

Author Response

The statement "In fact, the OR and 95% CI of definite intent and definite intent against vaccination are very close." was referring to the correlations between vaccine intent and concerned about affordability of the vaccine. Sorry if it was not made clear enough in the first review report. In table 3, the OR (95% CI) numbers for definite intent, definite or probable intention, and definite intent against vaccination are 1.51 (1.01-2.25), 1.30 (0.88-1.90), and 1.40 (0.72-2.74), respectively. Authors did explain the relationship between the affordability and vaccine intent in the response. However, given these close results, one should be more careful in stating that concerned about affordability predicts vaccine intent

Please note that although odds were close, only one of these was statistically significant- relationship between definite vaccination intent and concerns about vaccine affordability (OR: 1.51, 95% CI: 1.01 – 2.25).

However, we have noted your point and incorporated in our discussion.  We have added the following texts in discussion:  “Although majority of the people are willing to pay for a vaccine, we found that individuals concerned about vaccine affordability are more likely to have definite vaccination intent. Perhaps those do not have a definite intent does not care about vaccine affordability, but those have definite intent they care about vaccine affordability considering that the Bangladesh government may not be able to cover the huge population or might take years to cover the population. Therefore, participants with a definite intent might have to resort to private providers if the vaccines are available from them and here comes the concerns of affordability. However, given close results, we should be cautious in stating that concerns about affordability predict vaccine intent. Further studies are recommended to investigate this association”

The percentage of definite yes and definite no appear in abstract, results text and figure 2. The discrepancy between results text and figure 2 were revised, but the numbers in abstract are still different from the others. Authors are invited to make the results consistent across the whole manuscript.

Done. See the manuscript. (Highlighted in yellow)

Reviewer 2 Report

The papers has been improved following the suggestions from the previous review round. Anyway, there are still some points that would deserve a revision in my opinion:

  • Authors quoted "By the end of the data collection period, 700 individuals responded to our invitation, of which three refused participation. Hence, our
    final sample size included 697 individuals". I have some troubles in understanding this. I mean that Authors cannot known the number of people who refused to participate as social media platforms have been used to disseminate the survey. Saying that only three refused to participate is really misleading. Furthermore, it is not clear why these three persons responded to the invitation and then refused to participate. How did the Authors collect their response?
  • Methods should enclose the currency (and any exchange rate) used to quantify the WTP.
  • As the Authors decided to not stratify the analysis by previous COVID-19 infection, I would invite them to elaborate on this issue in discussion.

Author Response

Methods should enclose the currency (and any exchange rate) used to quantify the WTP.

We have collected data in the local currency, Bangladeshi Taka. In reporting, we converted the local currency, BDT, into United States Dollar (USD). One USD is equivalent to 84.8 BDT (https://www.bb.org.bd/econdata/exchangerate.php).

We have updated methods section with this information.

As the Authors decided to not stratify the analysis by previous COVID-19 infection, I would invite them to elaborate on this issue in discussion.

We have elaborated this issue in discussion (highlighted in yellow).  

Reviewer 3 Report

The authors answered all the questions appropriately and thoroughly. The manuscript has been improved a lot after revision. It can be accepted in present form. 

Author Response

Thanks again for your valuable comments and for accepting our manuscript.

Reviewer 4 Report

Thank you very much for changes. In my opinion, the paper may be accept in present form. Good job!

Author Response

(The authors gave the same response as above.)
